# Face Memorization Using AIM Model for Mobile Robot and Its Application to Name Calling Function

**DOI:** 10.3390/s20226629

**Published:** 2020-11-19

**Authors:** Masahiko Mikawa, Haolin Chen, Makoto Fujisawa

**Affiliations:** Faculty of Library, Information and Media Science, University of Tsukuba, 1-2 Kasuga, Tsukuba, Ibaraki 305-8550, Japan; sikon766@gmail.com (H.C.); fujis@slis.tsukuba.ac.jp (M.F.)

**Keywords:** social human–robot interaction, face memorization, face classification, user acceptance, AIM model, sleep function

## Abstract

We are developing a social mobile robot that has a name calling function using a face memorization system. It is said that it is an important function for a social robot to call to a person by her/his name, and the name calling can make a friendly impression of the robot on her/him. Our face memorization system has the following features: (1) When the robot detects a stranger, it stores her/his face images and name after getting her/his permission. (2) The robot can call to a person whose face it has memorized by her/his name. (3) The robot system has a sleep–wake function, and a face classifier is re-trained in a REM sleep state, or execution frequencies of information processes are reduced when it has nothing to do, for example, when there is no person around the robot. In this paper, we confirmed the performance of these functions and conducted an experiment to evaluate the impression of the name calling function with research participants. The experimental results revealed the validity and effectiveness of the proposed face memorization system.

## 1. Introduction

In recent years, social robots used in public spaces have been studied and developed actively. We are studying an information service robot with a face in a snowy cold region, as shown in Figure 1. When the robot wandered among pedestrians on a sidewalk, many children were glad to meet it. However, although the robot has a face with eyes and an exterior body made with soft material like a snowman, some adults obviously avoided the robot, and it seemed that the robot made them feel uncomfortable.

Thus, there have been several studies to investigate people’s impressions of a robot. Bartneck et al. investigated how robot design influences people [1]. Several emotion models for social robots and robot eye-gazing methods for human–robot interactions have been reviewed in [2,3] respectively. Influences of robot head movements were investigated in [4,5,6]. Huang et al. proposed the friendly behavior design method for interactive robots based on a combination of response time, approach speed, distance and attentiveness [7]. Shulte et al. developed a mobile robot with the mechanical face that could express emotion using face expressions and voice [8]. It was shown that speaking was effective at attracting people through experiments in [9].

Kanda et al. or Ishiguro et al. conducted human–robot interaction experiments in several environments: an elementary school, an office and a shopping mall [10,11,12,13]. The mobile robots used in these experiments could call a person’s name by using information embedded in the RFID tag, and it was described in [11] that, “Many children were attracted by the robot’s name-calling behavior.” Chen et al. [14] conducted experiments in which a robot called out to a person by name, and investigated the reaction time, and showed that there was a significant difference between calling out a person by name and not calling out a person by name. Like these studies, name calling is one of the important functions for social robots. However, the studies are still not enough, and there is no experiment that quantitatively evaluates the impression the robot gives to a person by name calling.

We studied a name calling function using a face memorization system. In order to build the face memorization system, face detection, face classification and classifier (re-)training functions, and speech recognizer and text-to-speech functions are required. Various methods for the face detection and the face classification, which are important functions in this study, have been proposed. Broadly classified, there are 2D image-based methods that use only features extracted from an image, and 3D based methods that use depth information obtained from a RGB-D camera or LiDAR in addition to image features. The 3D based methods such as [15,16,17] are more robust than the 2D image-based methods because they use depth information too. However, there are two problems: one is the cost of the device for measuring the depth information, and the other is the weakness of the device to sunlight and strong light because it uses infrared or laser light to measure the depth information. On the other hand, many 2D image-based methods have been proposed [18,19], and ordinary cameras can be used. Since it is difficult to say which is better at the present time, in this study, we adopted the 2D image-based method, which allows using an inexpensive, ordinary camera, for the robotic system shown in Figure 1.

In addition to the face detection and classification, it is necessary for the social robot to memorize the face and name of a stranger that the robot meets for the first time. However, it takes a long time to (re-)train the face classifier; moreover, the number of calculations is large. On the other hand, the interaction between a person and the robot is not always busy. For example, when there are no people around the robot, the robot has nothing to do.

Our robot has a sleep–wake function based on our proposed mathematical activation-input-modulation (AIM) model. Multiple processes for controlling the robot system run in parallel, and the AIM model transitions between sleep and wake states depending on the stimuli detected by an external sensor, and controls the frequency of execution of these processes. In the wake state, the face detection and classification functions work hard during detecting persons’ faces. When the robot detects a stranger, it stores her/his face images with her/his name after getting her/his permission. When there is no person around the robot, it begins sleeping through the operation of the AIM model. There are two sleep states: rapid eye movement (REM) sleep and non-REM sleep. The face classifier is re-trained by adding newly collected face images in the REM sleep state just like a person dreams, and the execution frequencies of almost all of information processes are reduced in the non-REM sleep state. This means that the operation of the AIM model can decrease battery draining, and as the result, the operating time of the mobile robot can be increased.

In summary, the features of our social robot system are that it has the function to call a person by her/his name and the function to (re-)train a stranger’s face and name during the sleep state controlled by the mathematical AIM model. The face memorization system controlled by the sleep function is shown in Section 2. The experimental system with the face memorization function and the effects of the function for the mobile robot system are shown in Section 3. Finally, in Section 4, we describe the results of an experiment to evaluate the impression of the name calling function on research participants, and discuss them.

## 2. Face Memorization System for a Mobile Robot

The basic idea of the face memorization system has been proposed in our previous work [20]. Although there are some overlaps, since there are some modifications and additions, the detailed system is described in this section.

### 2.1. System Outline

An outline of our proposed face memorization system is shown in Figure 2. First, when a mobile robot detects a person, it identifies whether it is a stranger or not based on a face classifier. When the robot classifies her/him as a stranger, it asks permission for collecting her/his facial images and name. Second, it is necessary to re-train the face classifier by adding newly collected face images and names of strangers, but the computational cost for re-training is high and it takes much time to re-train it. Thus, our proposed system executes the re-training process when there is no person around the robot. Multiple processes are executed in parallel in the system, and their behaviors are controlled by the activation-input-modulation (AIM) model described in the following subsection. Since it is unnecessary to process external sensor data in real-time when there is no person around the robot, the robot sleeps depending on a state of the AIM model. Specifically, the execution frequencies of processes for external sensors decrease, and the re-training process begins to be executed just as though the robot dreams. Last, when the robot detects a person again, it wakes up by the operation of the AIM model. In other words, the face classification process begins to be executed again. When the robot detects a known person, it can call her/him by her/his name.

This system described above is controlled by a finite state machine shown in Figure 3. The system has four states: awaken, relaxed, non-REM sleep and REM sleep. The state transitions are described in detail in the next sub-section. While the robot detects a person, the system is in the waking state. The robot calls the name of her/him it remembers, or collects her/his facial images when she/he is a stranger. After the robot detects no person, the system shifts to the sleep state through the relaxed state. In the REM sleep state, the face classifier is (re-)trained.

### 2.2. Mathematical AIM Model

The AIM state-space model proposed by Hobson [21] is shown in Figure 4. Humans’ consciousness states are expressed depending on levels of the following three elements. Activation controls processing power. Input switches information sources. Modulation switches external and internal information processing modes. In an waking state, external information obtained by external sensory organs is processed actively. A human dreams in rapid eye movement (REM) sleep, and it is said that stored internal information is processed actively. In non-REM sleep, input and output gates are closed and the processing power declines totally.

We have designed a mathematical AIM model [22,23] for controlling sensory information processing systems of an intelligent robot based on the AIM state space model. Since the details of the mathematical AIM model have been described in these previous works, only the summary is explained in this paper.

The block diagram of the mathematical AIM model is shown in Figure 5. The mathematical AIM model controls ratios among the external and internal information processors. The element *S* calculates stimuli, such as changes of perceptual information, extracted from sampled data. The element *A* decides an execution frequency of each information processor. The element *I* decides parameters used when stimuli are calculated in *S*. The element *M* decides an execution frequency for each sensory data capture. Each element consists of two sub-elements. The subscripts ex and in mean that the sub-elements relate to external and internal respectively.

An example of the variations of the sub-elements with time is shown in the upper figure of Figure 6. While an external stimulus is detected, the levels of the elements related to the external are higher than those related to the internal. After the external stimulus becomes lower than a threshold, the state is shifted to relaxing. After a certain time, first, the state is shifted to the non-REM sleep. Then, each level increases and decreases periodically. When again a stimulus is detected, the state shifts to the waking state. The lower figure of Figure 6 shows the variations of *A*, *I* and *M* calculated from their sub-elements with time.

### 2.3. Processes for Face Memorization Function Controlled
by the AIM Model

Figure 7 shows relations between the mathematical AIM model and processes for the face memorization function. A microphone is used for collecting persons’ names by voice recognition. A camera is used to detect people’s faces, and collect images of strangers’ faces. When a stranger’s face is detected, the robot asks for her/his name using text-to-speech function and collects her/his face images and name with her/his permission.

The face memorization function consists of external information processing and internal information processing. In the external information processing system, image data that require real-time processing are handled. In the internal information processing system, the following kinds of data are treated: When the data are difficult to process in real-time, (taking too much time). When it is not necessary to process in real-time.

The external information processing system consists of the image capture, the face detection, the face image collection and the face classification functions. When this system detects a person’s face, it classifies whether she/he is a stranger or not. When she/he is a stranger, the robot asks for her/his name, and at the same time collects her/his face images. When she/he is an acquaintance, since the robot knows her/his name based on the classifier, the robot can call her/him by name. Since it is necessary to execute these processes in real-time, they are executed in the waking state.

The internal information processing system has functions related to memories, and consists of a function for generating training images from both the face images and their names newly stored in the waking state and a function for (re-)training the face classifier using the generated training images of faces. It is difficult to execute both the training face image generation and face classifier (re-)training in real-time.

An execution frequency of each process is controlled depending on a value of the sub-element of the AIM model shown in Figure 6. The execution frequencies of the face detection and classification are decided based on the value of a_ex. For example, when a_ex becomes higher, the external information processing frequency increases. The frequency of the external sensory data capture is decided based on m_ex. The training image generation and the face classifier training are executed depending on a_in.

### 2.4. Face Memorization

The face memorization system mainly consists of the following five functions. (1) A face detection function is used to detect people’s faces and crop and collect face images. (2) A face classification function can estimate class probabilities based on a trained model. (3) A face classifier training function is used to re-train the face classifier with previously the collected face images, newly added face images and the people’s names. (4) A speech recognition function is used for simple conversation and to collect people’s names. (5) A text-to-speech function is used to have a conversation and call a person by her/his name.

In order to crop a face area from a captured image, the multi-task cascaded convolutional neural network (multi-task CNN) proposed by Zhang et al. [24] is used in this paper. Although we tried the Haar Cascades classifier in OpenCV and the Face Detector in Dlib, the multi-task CNN is better than these two detectors regarding computational time, detection rate for an image with a complex background and robustness for changes of face direction or face brightness.

The FaceNet proposed by Schroff et al. [18] and the FaceNet model pre-trained by Sandberg [25] are used to extract a 512-dimensional feature vector for a face image. The support vector machine algorithm in scikit-learn [26] is used to implement a face classifier.

Julius [27] and Open JTalk [28] are used for Japanese speech recognition and text-to-speech respectively.

### 2.5. Consciousness State Transition

The current state of the robot system is determined depending on the combination of the values of *A*, *I* and *M* as shown in Table 1. Moreover, there are two kinds of modes in the waking state. One is a face image collecting mode, and the other is a face classification mode. The system is normally in the face classification mode. When the face classifier finds a stranger’s face, the mode translates to the face image collecting mode. Figure 8 shows both a state transition diagram and values of execution frequencies related to the image processing. The execution frequency of the face detection does not become less than 1 (fps) even in the REM or non-REM sleep; as a result, the system can wake up from sleep when a human face is detected in a captured image.

## 3. Experimental System

### 3.1. System Configuration

Figure 9 shows the mobile robot system. The wheeled mobile robot, Pioneer 3-DX (OMRON Adept Technologies, LLC., Pleasanton, CA, USA), was equipped with a USB camera and a tablet PC. A note PC (Ubuntu 16.04, Intel Core i7-2630QM, Memory 8 GB) for controlling the system was put on the mobile robot. The PC was supplied electric power from internal batteries of the mobile robot through a DC/AC converter.

A robotic face with two eyes was displayed on the tablet PC; the direction of eyes often changed randomly in a horizontal direction, and the eyes sometimes blinked.

Software libraries are shown in Table 2. Robot Operating System (ROS) [29] is used for communications among processes, TensorFlow [30] is used for detecting faces and models of face recognition, scikit-learn [26] is used for training the face classifier, OpenCV [31] is used for several image processing tasks, Julius [32] is used for speech recognition and Open JTalk [28] is a Japanese text-to-speech system.

### 3.2. Implementation of the Face Classifier

Figure 10a–c shows some face classification results. As shown in Figure 10a, the face classifier works well when a face is rotated obliquely. As shown in Figure 10b,c, there was no influence of the presence or absence of a pair of glasses, and a person who was not included in the trained classifier could be classified as a stranger. Table 3 shows the computational times of the face detection and the classification.

As described in the Section 2.4, the face image (re-)training function consists of the prepossessing where face images are cropped and stored by the multi-task CNN, the feature vector transformation by the FaceNet and the (re-)training by the SVM. Table 4 shows execution speeds of the prepossessing, the feature vector transformation and the re-training with or without the mathematical AIM. In total, 7000 face images of 35 persons were collected for these calculations. Of those, 3500 face images of 35 persons were used for re-training the face classifier, and the rest were used for evaluating the re-trained face classifier.

By using the AIM model, since the system changes its states between the REM sleep where these processes related to the re-training are executed and the non-REM sleep where most processes do not work periodically, it takes longer to finish re-training the classifier than the calculation without the AIM model. However, we believe that it is not problem, because the re-training is executed while the system has nothing to do and sleeps. The number of persons for the classification evaluation was 35 in this paper. Although 35 was not so many, the classification results could achieve 100%.

### 3.3. Effectiveness of AIM Model

It is important for a mobile robot to save its battery. In order to show another effectiveness of the AIM model, we conducted a simple experiment wherein the only executed processes were the face detection and the face classification. However, the computational loads of these processes are high, and the execution frequency was about 4.8 (fps) in the experimental system, as described in Section 3.1. The variations of battery voltage with time were measured under two conditions: with the AIM model and without the AIM model.

Figure 11 shows the battery voltage variations with time with/without AIM. In the case of without the AIM model, since the face detection and classification processes were executed as fast as possible, the voltage decreased quickly. In the case of with the AIM model, although we woke the system by showing our faces in front of the camera about every five minutes, since the execution frequency of the image processing became lower through the operation of the AIM model in the sleeping mode, the electricity consumption could be cut.

## 4. Experimental Results

Previous studies have shown that a robot’s behaviors can affect the impression that a person gets from the robot. Traeger et al. reported that behaviors of a social robot influenced not only the communication between the robot and humans, but also the communication between humans [33]. When a robot and a person pass each other in a corridor, the experiments in [34] have confirmed that the impression that the robot’s actions give to the person is different depending on whether the direction of the robot’s face and gaze is controlled or not.

Therefore, various factors may affect the experiments under realistic experimental conditions. Thus, in order to properly investigate the effectiveness of the name calling function using the face memorization system, we chose a simple experimental environment to evaluate research participants’ impressions.

### 4.1. Outline of Experiments

In order that the mobile robot have the ability to call a person by her/his name, we developed the face memorization system. We conducted some experiments under the following two conditions:(1)The robot calls a research participant using her/his name during a conversation.(2)The robot calls the participant using the pronoun “you” during a conversation.

That is to say, the purpose of the experiments was to investigate what kind of impression the robot would give a person by using her/his name.

In this experiment, we chose the simplified Old Maid as the card game to play during a conversation. The situation and the rules are simple: the research participant has two trump cards, among which the joker is included, and the robot estimates which card the joker is. However, if an experimenter explains to the participants the purpose of these evaluation experiments in advance, the participants pay attention the robot’s way of addressing them too much, and as a result, they may not be able to give fair evaluations. Thus, in this study, before the experiments, the experimenter gave the participant a false explanation as follows:(a)The robot has a face memorization function.(b)The robot has face expression detection and analysis functions, and has an ability to estimate the position of the joker based on a human’s facial expressions.(c)The purpose of this experiment is not to win Old Maid, but to let the robot guess which of two cards is the joker correctly.

After all the experiments, the experimenter told the participants the true experimental purpose.

The research participant had a meeting with the robot three times in a set of evaluation experiments. On the first meeting, the participant’s name and face images were collected by the robot. Upon the second and third meetings, she/he played the card game with the robot and evaluated her/his impression of interaction with the robot after each meeting. The robot used the participant’s name during one meeting, and it called them by the pronoun “you” during the other meeting. The experiments were conducted with a counterbalanced measures design.

### 4.2. Procedure of Experiments

The detailed procedures of the experiments are described below.
(1)An experimenter explains the fake experiment’s purpose to a research participant in a waiting space, and takes her/him to an experimental space shown in Figure 12 described in the following subsection.(2)The mobile robot that stands by in the experimental space says hello to the participant.(3)The robot asks permission to collect her/his name and face images for experimental records.(4)After the robot finishes collecting the face images, the participant returns to the waiting space once.(5)The experimenter explains to the participant that she/he has the second and third meetings with the robot and plays the card game six times in one meeting after this explanation. Then the participant enters the experimental space.(6)After sitting in a chair, she/he shuffles two cards and sets them in a card holder on a table. The robot answers which of two cards is the joker. This transaction is repeated six times in one meeting.(7)After the second meeting, the participant returns the waiting space again, and answers a questionnaire given by the experimenter.(8)The procedure (6) is repeated once more as the third meeting.(9)After the third meeting, the participant returns the waiting space again, and answers another questionnaire given by the experimenter.(10)The experimenter tells the subject the true purpose of the experiments.

The detailed dialogue between a participant and the robot is shown in Section A.1 and Section A.2.

Here, in order to evaluate participants’ impressions properly according to the purpose of this study, these experiments were conducted under the following conditions. Basically, we wanted to avoid bad influences because of failures of the image or voice recognition.
(i)All the experiments were conducted based on the Wizard of Oz (WOZ) method [35]; the mobile robot was teleoperated by a human operator.(ii)The robot could get the correct position of the joker five out of six times but not the fourth time. This was realized by using a hidden camera.

Each participant evaluated her/his impression using the Godspeed Questionnaire [36]. It is said that, “The Godspeed Questionnaire Series (GQS) is one of the most frequently used questionnaires in the field of human–robot interaction” [37]. Table 5 shows the application of the five Godspeed questionnaires with a 5-point Likert-scale.

### 4.3. Experimental Environment

Figure 12 and Figure 13a,b show an experimental environment. The waiting space and the experimental space were divided by a partition. Only one research participant entered this place at once. The experimenter always sat on a chair in the waiting space. An operator for the WOZ was in a room next door, and teleoperated the mobile robot in the experimental space with a gamepad and a monitor. The operator could watch the entire experimental room and the position of the joker set on the card stand through a hidden camera shown in Figure 13a. Figure 13b shows one scene during an experiment.

### 4.4. Evaluation Results

A total of 22 participants in their twenties were recruited in this evaluation experiment. The number of females was 10, the number of males was 12 and the average age was 22.1 years old. They were all undergraduate or graduate students in the University of Tsukuba, Japan, and some were international students.

The standard deviations of the results of the impression evaluation using the Godspeed Questionnaire are shown in Figure 14, and the results of t-tests are shown in Table 6. Although all the averages under conditions where the robot called participants by their names were higher than those under conditions wherein it called them by the pronoun “you”, there was a significant difference in only the likeability. It can be said that the robot could give a better impression to a person by calling her/him by her/his name during conversation.

### 4.5. Discussion

We could confirm the basic validity and effectiveness of the name calling function using our proposed face memorization system through the experiments with research participants.

At the same time, the following facts were revealed from the interviews with the research participants or free descriptions in the questionnaire after the third meeting in the experiment.
(1)Some participants did not notice the difference between the second and third meeting with the robot.(2)Some participants believed that the robot really had the ability to estimate the position of the joker based on human facial expressions. Since the robot could estimate correctly five out of six times, they had the impression that this robot’s ability was awesome.(3)Since the way of the robot’s speaking using the text-to-speech software was polite and humble, some participants had good impressions of the robot.(4)The two CG eyes of the robot shown in Figure 9 sometimes blinked and the line of sight was changed. However, since the movements were not synchronized with the conversation, there were some research participants who answered the questionnaire by saying that “it was impersonal”.

Including these points, the weaknesses of this paper are shown below.
(a)Despite the simple experimental environment, the contents of the experiment and the behaviors and appearance of the robot might have some influence on the impression evaluation.(b)The experiments were conducted based on the WOZ in order to avoid malfunctions in image processing and voice recognition.(c)The number of persons whose faces were memorized by robots was still small.

Thus, in the future work, we believe that it is necessary to conduct more experiments under the following two types of environments. One is an experimental environment that is simple but less affected by the appearance of the robot and the tasks, and the other is a more realistic experimental environment for evaluating the usefulness of the entire system. In addition, since the progress of face detection and face classification is moving rapidly, it is necessary to always adopt the latest algorithm in order to improve the performance of the system.

## 5. Conclusions

We proposed a face memorization system with a sleep–wake function for a social mobile robot, and applied it to a name calling function. When the robot meets a stranger, although it stores her/his face images and name in storage, since it takes long time to re-train a face classifier, the re-training of the classifier is not executed immediately, but is executed in a sleep state where there is no person around the robot and there is no sensory information to process. We conducted two kinds of experiments in this paper. One was that battery voltage variations with time were measured with/without the sleep–wake function by the AIM model. Although the experimental conditions were simple, the battery running time using with the AIM model was a few times longer than that without the AIM model. The other was that we conducted an impression evaluation of the name calling function with research participants. Research participants evaluated their impressions of the robot that called them by their names through conversation during a simple card game. The Godspeed Questionnaire was used for the impression evaluation; it was confirmed that there was a significant difference in only the likeability. It can be said that the robot could give a better impression to a person by calling him/her by her/his name during conversation. The experimental results revealed the validity and effectiveness of the proposed face memorization system.

## Figures and Tables

**Figure 1 sensors-20-06629-f001:**
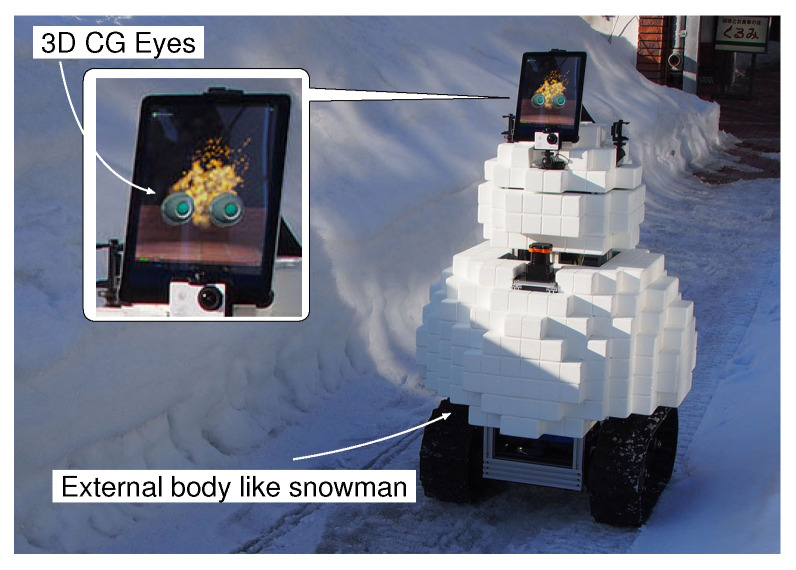
Information service robot on a sidewalk.

**Figure 2 sensors-20-06629-f002:**
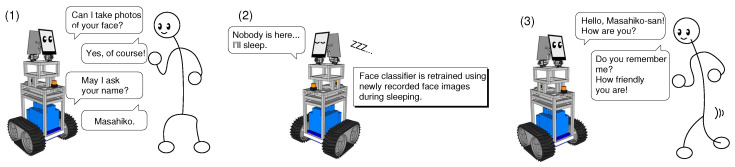
Calling name function based on face memorization.

**Figure 3 sensors-20-06629-f003:**
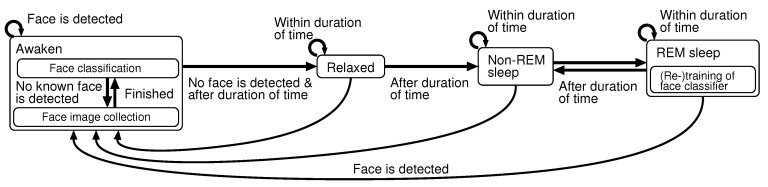
Finite state machine of system.

**Figure 4 sensors-20-06629-f004:**
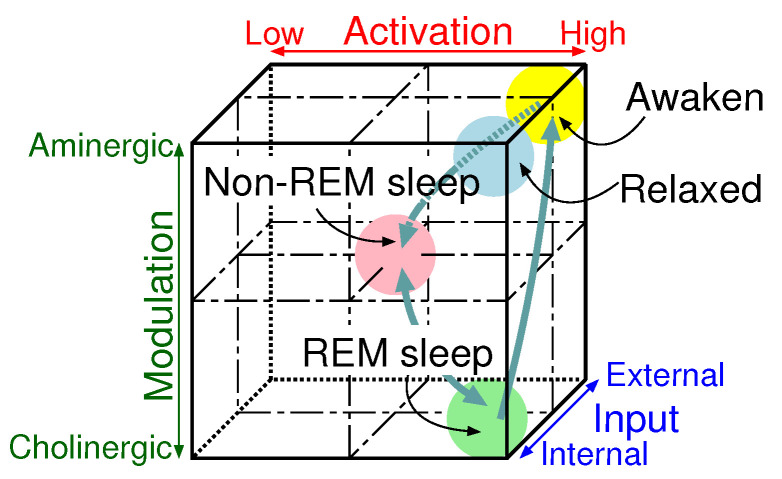
AIM model.

**Figure 5 sensors-20-06629-f005:**
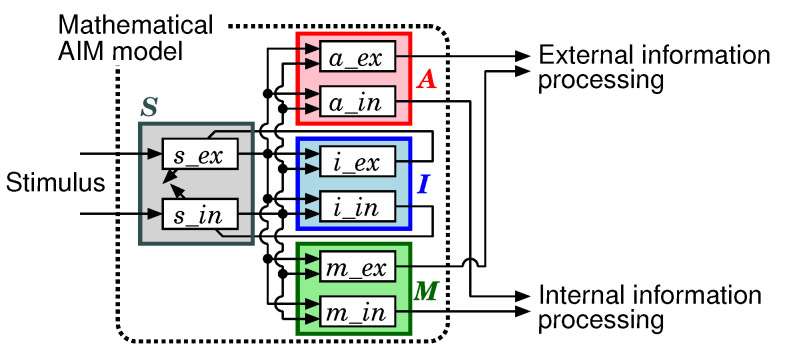
Mathematical AIM model.

**Figure 6 sensors-20-06629-f006:**
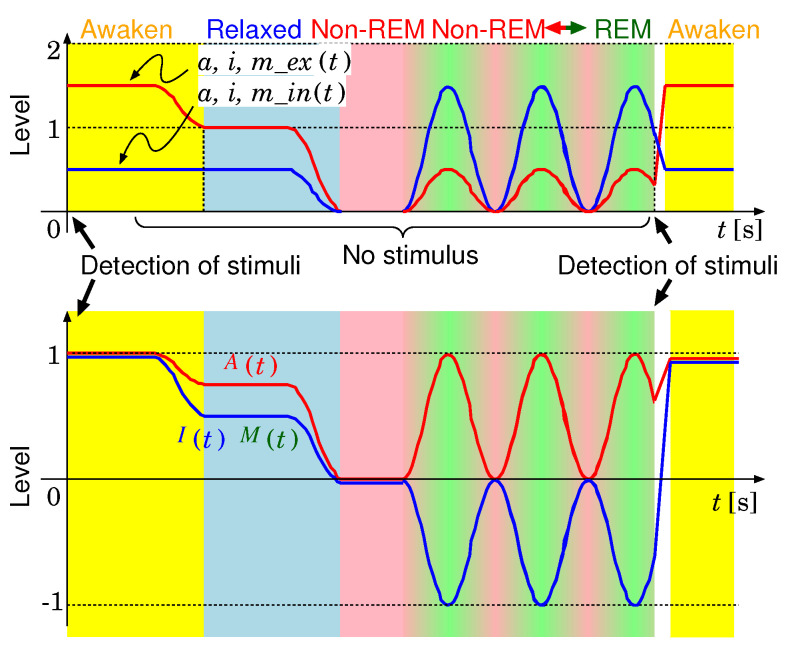
Variations of *A*, *I*, *M* and their sub-elements with time.

**Figure 7 sensors-20-06629-f007:**
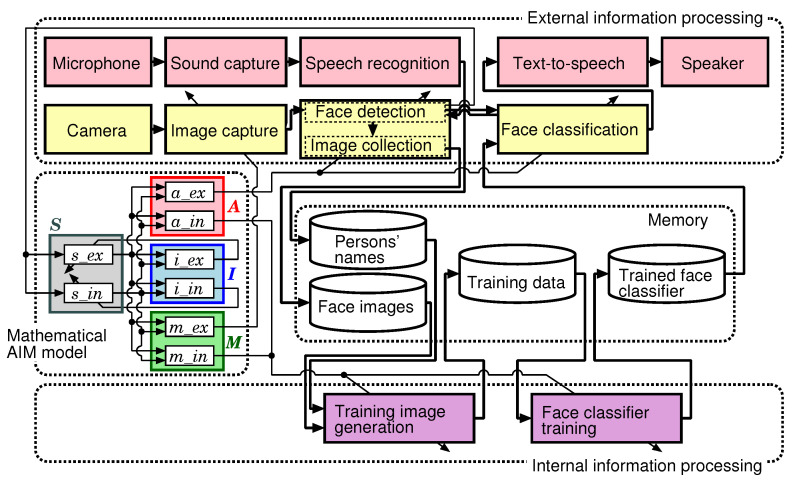
Block diagram of the AIM model and processes for face memorization function.

**Figure 8 sensors-20-06629-f008:**
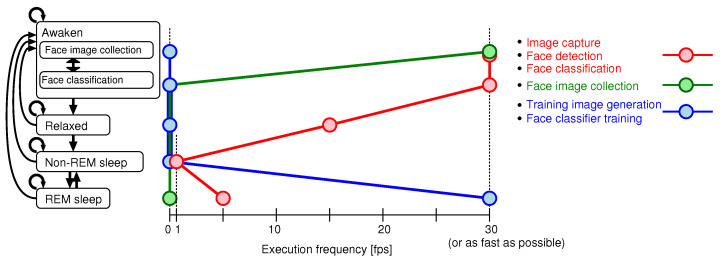
Execution frequency of each process.

**Figure 9 sensors-20-06629-f009:**
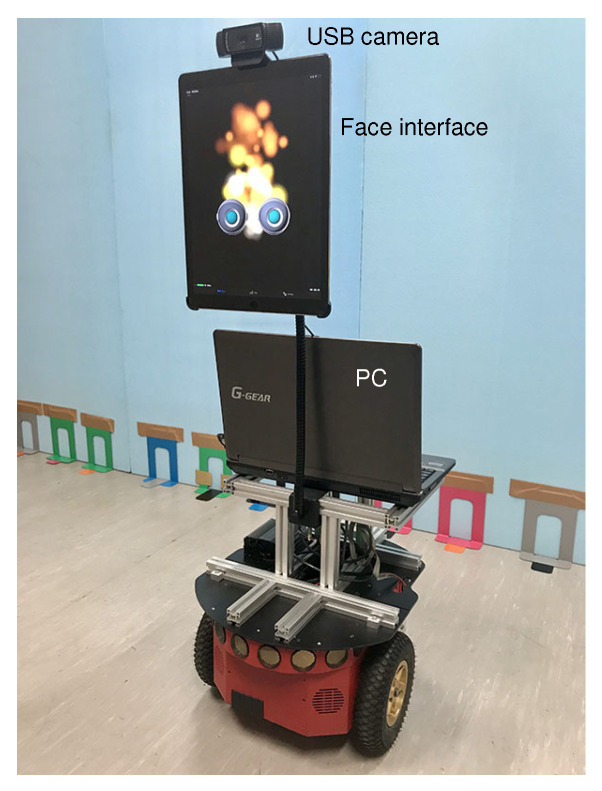
Mobile robot.

**Figure 10 sensors-20-06629-f010:**
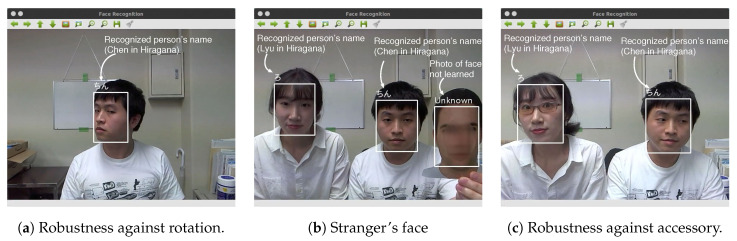
Face classification examples.

**Figure 11 sensors-20-06629-f011:**
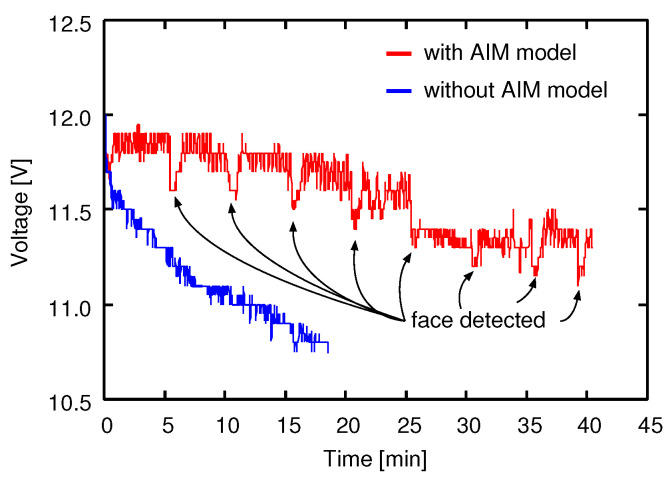
Variations of battery voltage with time.

**Figure 12 sensors-20-06629-f012:**
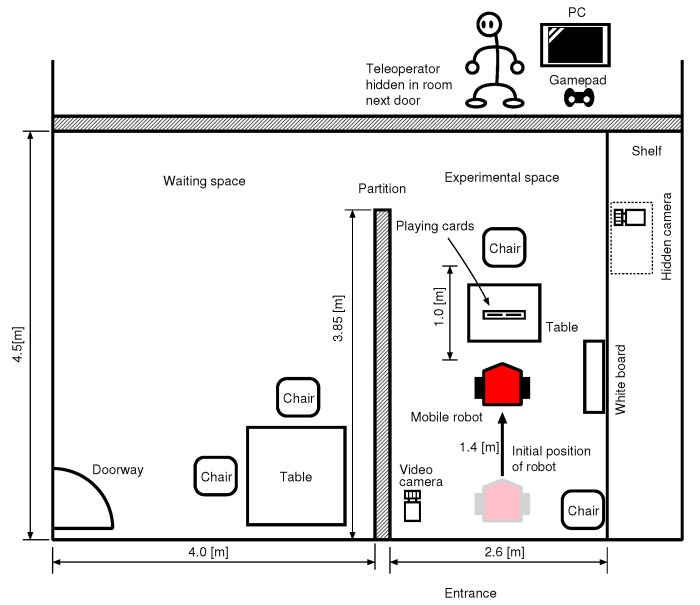
Layout of experimental environment.

**Figure 13 sensors-20-06629-f013:**
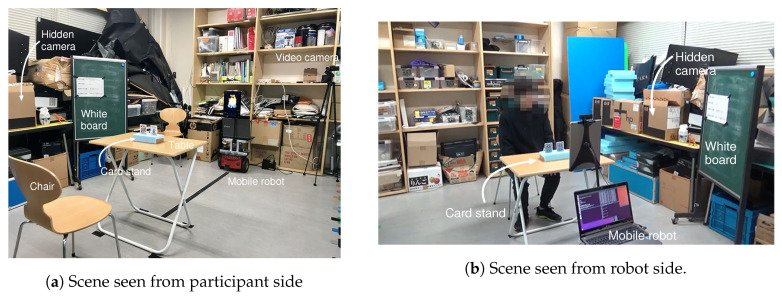
Scene example of evaluation experiment.

**Figure 14 sensors-20-06629-f014:**
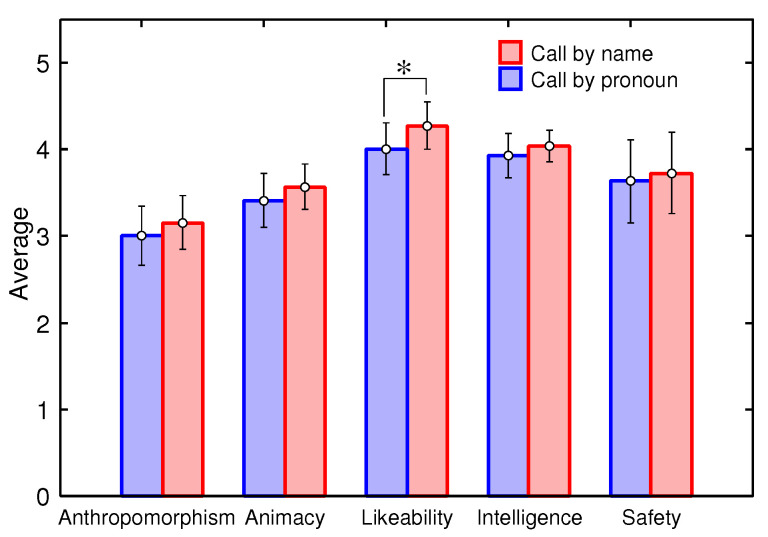
Evaluation results.

**Table 1 sensors-20-06629-t001:** Relations among AIM values and consciousness states.

State	Value of *A*	Value of *I*	Value of *M*
Awaken	0.75<A≤1.0	0.5<I≤1.0	0.5<M≤1.0
Relaxed	0.5<A≤0.75	0.0<I≤0.5	0.0<M≤0.5
Non-REM	0.0<A≤0.5	−0.5<I≤0.0	−0.5<M≤<0.0
REM	0.5≤A≤1.0	−1.0≤I≤−0.5	−1.0≤M≤−0.5

**Table 2 sensors-20-06629-t002:** Versions of software libraries.

Library	ROS	TensorFlow	Scikit-Learn	OpenCV	Julius	Open JTalk
Version	Kinetic Kame	1.12.0	0.20.1	3.3.1	4.3.1	1.07

**Table 3 sensors-20-06629-t003:** Computational times of face detection and face classification.

	Computational Time
Face detection	0.14 [s]
Face classification	0.07 [s]
Face detection & classification	0.21 [s] (=4.77 [fps])

**Table 4 sensors-20-06629-t004:** Computational time for re-training face images.

	With AIM	Without AIM
Prepossessing	1589.4 [s]	1040.1 [s]
FaceNet	791.7 [s]	356.1 [s]
SVM	7.88 [s]	7.23 [s]
Precision	100.0 [%]	100.0 [%]

**Table 5 sensors-20-06629-t005:** The Godspeed Questionnaire Series.

Section	Items		
Anthropomorphism	Fake	-	Natural
	Machinelike	-	Humanlike
	Unconscious	-	Conscious
	Artificial	-	Lifelike
	Moving rigidly	-	Moving elegant
Animacy	Dead	-	Alive
	Stagnant	-	Lively
	Mechanical	-	Organic
	Artificial	-	Lifelike
	Inert	-	Interactive
	Apathetic	-	Responsive
Likeability	Dislike	-	Like
	Unfriendly	-	Friendly
	Unkind	-	Kind
	Unpleasant	-	Pleasant
	Awful	-	Nice
Perceived Intelligence	Incompetent	-	Competent
	Ignorant	-	Knowledgeable
	Irresponsible	-	Responsible
	Unintelligent	-	Intelligent
	Foolish	-	Sensible
Perceived Safety	Anxious	-	Relaxed
	Agitated	-	Calm
	Quiescent	-	Surprised

**Table 6 sensors-20-06629-t006:** Evaluation results.

Scale		Call by Name	Call by Pronoun
Anthropomorphism	Mean	3.155	3.009
	Standard deviation	0.617	0.683
	Significance probability	0.195	
Animacy	Mean	3.568	3.409
	Standard deviation	0.521	0.621
	Significance probability	0.226	
Likeability	Mean	4.273	4.009
	Standard deviation	0.544	0.606
	Significance probability	0.039 (*)	
Perceived Intelligence	Mean	4.036	3.927
	Standard deviation	0.368	0.511
	Significance probability	0.117	
Perceived Safety	Mean	3.727	3.636
	Standard deviation	0.935	0.959
	Significance probability	0.701	

* *p* < 0.05.

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
