# Peer review of "Face Memorization Using AIM Model for Mobile Robot and Its Application to Name Calling Function"

_sensors, 2020, doi:10.3390/s20226629_

Round 1

Reviewer 1 Report

This paper proposes a face recognition method for social mobile robots.

The algorithms presented in the manuscript are not new and a trivial combination of existing methods. 

However, the robot system and hardware are somewhat new and well developed. 

Because the manuscript contributes the development of a new robot embedded system, it is recommended to report  a whole computational time. 

It would be better to evaluate the system under various real-world environments. 

Author Response

All the review reports for three reviewers are collected in one pdf file.

The Reviewer 1, please refer to pages 1-4 of the attachment.

Reviewer 2 Report

The authors developed a social mobile robot that has a name calling function using a face memorization system. It is an interesting topic. The experimental results revealed the validity and effectiveness of the proposed face memorization system. However, there are some issues should be discussed as follows.

  1. This manuscript looks like a technique report but not a paper.
  2. The innovation of this manuscript should be emphysized in the introduction part.
  3. Too much description of the system construction are given in each part of the text.
  4. There are no experimental comparison with other references, especially in the state of the arts.

Author Response

All the review reports for three reviewers are collected in one pdf file.

The Reviewer 2, please refer to pages 5-9 of the attachment.

Reviewer 3 Report

The paper deals with an interesting topic face analysis for robotic puropse. In the present form the paper needs improvements. Starting from the introduction authors should provide a wider picture about the reserch domain analysed introducing some of the last reserches done on face analysis also by the use of 3D sensors rather than only 2D ones, that seems to be less reliable. Some more references should be added as for instance the following ones:

Dagnes, N., et al. (2018). 3D geometry-based automatic landmark localization in presence of facial occlusions. Multimedia Tools and Applications77(11), 14177-14205. 

Regarding the methodological section authors should provide a fisrt section in which the proposed approach is proposed from a global point of view for goingo further in the specific methodological stages. The introduction of a graphical flowchart, reagrding the overall methodology, could support its readibility. Reagding the experimental validation more details regarding the experimentak setting should be necessary in order to understnad not only the methodological strenghts but also its weackness, in order to provide a more reilable added value to the scientific comunuty 

Author Response

All the review reports for three reviewers are collected in one pdf file.

The Reviewer 3, please refer to pages 10-14 of the attachment.

Round 2

Reviewer 3 Report

Authors have improved the scientific level of the paper